# Cosine-Based Embedding for Completing Lightweight Schematic Knowledge in DL-Lite$_{core}$ †

**Weizhuo Li** [1,2,3], **Xianda Zheng** [4,5,*], **Huan Gao** [6], **Qiu Ji** [1] **and Guilin Qi** [2,5]

1  School of Modern Posts, Nanjing University of Posts and Telecommunications, Nanjing 210003, China
2  Key Laboratory of Computer Network and Information Integration, Southeast University, Ministry of Education, Nanjing 211189, China
3  State Key Laboratory for Novel Software Technology, Nanjing University, Nanjing 210093, China
4  School of Computer Science, The University of Auckland, Auckland 1010, New Zealand
5  School of Computer Science and Engineering, Southeast University, Nanjing 211189, China
6  Intel Joint Research Institute on Intelligent Edge Computing, Nanjing 211135, China
*  Correspondence: zhengxianda@seu.edu.cn; Tel.: +86-185-0556-5540
†  This Paper is a Substantial Extended Version of Paper Published in the 8th CCF International Conference on Natural Language Processing and Chinese Computing, Dunhuang, China, 9–14 October 2019.

**Abstract:** Schematic knowledge, an important component of knowledge graphs (KGs), defines a rich set of logical axioms based on concepts and relations to support knowledge integration, reasoning, and heterogeneity elimination over KGs. Although several KGs consist of lots of factual knowledge, their schematic knowledge (e.g., *subclassOf* axioms, *disjointWith* axioms) is far from complete. Currently, existing KG embedding methods for completing schematic knowledge still suffer from two limitations. Firstly, existing embedding methods designed to encode factual knowledge pay little attention to the completion of schematic knowledge (e.g., axioms). Secondly, several methods try to preserve logical properties of relations for completing schematic knowledge, but they cannot simultaneously preserve the transitivity (e.g., *subclassOf*) and symmetry (e.g., *disjointWith*) of axioms well. To solve these issues, we propose a cosine-based embedding method named CosE tailored for completing lightweight schematic knowledge in DL-Lite$_{core}$. Precisely, the concepts in axioms will be encoded into two semantic spaces defined in CosE. One is called angle-based semantic space, which is employed to preserve the transitivity or symmetry of relations in axioms. The other one is defined as translation-based semantic space that is used to measure the confidence of each axiom. We design two types of score functions for these two semantic spaces, so as to sufficiently learn the vector representations of concepts. Moreover, we propose a novel negative sampling strategy based on the mutual exclusion between *subclassOf* and *disjointWith*. In this way, concepts can obtain better vector representations for schematic knowledge completion. We implement our method and verify it on four standard datasets generated by real ontologies. Experiments show that CosE can obtain better results than existing models and keep the logical properties of relations for transitivity and symmetry simultaneously.

**Keywords:** schematic knowledge; embedding; lightweight ontology; transitivity; symmetry

## 1. Introduction

In recent years, knowledge graphs (KGs) have attracted lots of attention since they can effectively organize and represent knowledge from rich resource data, which can provide users with various smarter services through knowledge reasoning techniques. There exist two types of knowledge in a KG. One is schematic knowledge, which is made up of assertions about concepts and relations called axioms. The other is factual knowledge, which is composed of statements about instances called triples [1].

Schematic knowledge, a critical component of KGs, formulates a rich set of logical axioms based on concepts to support the elimination of heterogeneity, integration, and reasoning over KGs. Nevertheless, existing knowledge graphs (e.g., WordNet [2], DBpedia [3],

and YAGO [4]) mostly consists of lots of factual knowledge and little schematic knowledge. For the famous knowledge graph DBpedia (https://www.dbpedia.org/ accessed on 5 September 2022), it contains more than 1.89 billion triples and over 3.5 million entities among them. However, it contains only 768 concepts and 20 *disjointWith* axioms asserted among them. The sparsity of schematic knowledge will limit the applications and services of KGs such as query-answering [5], recommendation system [6], and knowledge integration [7]. Hence, it is essential to improve the completeness of schematic knowledge. Nevertheless, it is hard for traditional reasoning-based methods to automatically infer all the remaining axioms. Take, for an example shown in Figure 1, there are three axioms (Farm_Boy, *subclassOf*, Boy), (Boy, *subclassOf*, Male_Person), and (Male_Person, *subclassOf*, Person), defined in one schematic knowledge. If the relation *subclassOf* from *Boy* to *Male_Person* marked in red is missing, then it is hard to obtain the *subclassOf* relation from *Boy* to *Person* marked in blue using traditional reasoning-based methods.

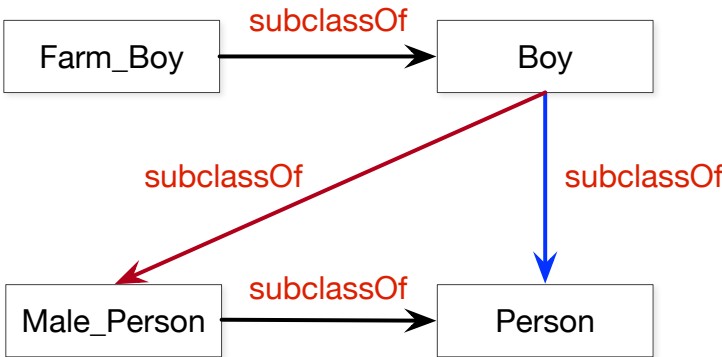

**Figure 1.** An example of the missing one *subclassOf* axiom for schematic knowledge completion.

　　Knowledge graph embedding, which aims to encode the entities and relations of a KG into the low-dimensional and continuous vector space, has been widely studied and has been proven to be of great help in KG completion via link prediction [8] and other downstream tasks; see [9–12]. The mainstream models are designed for factual knowledge embedding, including TransE [13], TransH [14], TransR [15], and so on, which regard the relation as a "translation" from the head entity to the tail entity. Another kind of model for factual knowledge embedding, such as RESCAL [16], DistMult [17], HolE [18], and ComplEx [19], design various operators to encode rich interactions among embedding vectors. Recently, several works have tried to encode the properties of relations for completing schematic knowledge, such as EmbedS [20], TransC [21], HAKE [22], RotatE [23], EL Embeddings [24], and OWL2Vec* [25]. Most of them have attempted to encode the concepts and instances into a spherical semantic space so that the transitivity and other logical properties of relations could be preserved. Benefiting from this idea, more potential axioms and triples could be predicted.

　　Although the methods of KG embedding have achieved great success in KG completion, most of them suffer from two limitations. On the one hand, the mainstream models of KG embedding mainly consider the triples derived from factual knowledge, but few of them pay attention to the modeling of the logical properties of relations. Hence, these methods can hardly be applied to the related tasks of schematic knowledge (e.g., completion, reasoning, and repairing). For example, given one axiom $(C_i, r, C_j)$ with two concepts $C_i, C_j$, and a symmetry relation $r$, for the translation-based KG embedding method TransE [13], if two concepts and a relation are projected into its defined semantic space, the axiom's score of $||\mathbf{C_i} + \mathbf{r} - \mathbf{C_j}||_2$ is not equal to $||\mathbf{C_j} + \mathbf{r} - \mathbf{C_i}||_2$. Therefore, the symmetry of relation $r$ is lost in the defined semantic space of TransE. On the other hand, existing embedding models for schematic knowledge mainly focus on modeling the properties of relations such as transitivity, (anti)-symmetry, inversion, and composition. It remains a challenge for them to simultaneously preserve the transitivity (e.g., *subclassOf*) and symmetry (e.g., *disjointWith*) of axioms well. As the schematic knowledge of KG usually has its logical foundations based

on ontology languages, such as the Resource Description Framework (Schema) (RDF(S)) (https://www.w3.org/TR/rdf-schema/ accessed on 5 September 2022) and the Ontology Web Language (OWL) (https://www.w3.org/OWL/ accessed on 5 September 2022), so it is important to improve the completion of *subclassOf* axioms and *disjointWith* ones, because both of them are basic axioms asserted in schematic knowledge or ontologies, which can ensure the quality of KGs and infer more implicit knowledge.

To solve the problems, we attempt to propose a cosine-based embedding method, namely CosE (**Cos**ine-based **E**mbedding), for learning the vector representations of concepts in lightweight schematic knowledge that corresponds to the ontology expressed in DL-Lite$_{core}$. DL-Lite$_{core}$ is a lightweight language of Description Logics (DL), which can capture basic ontology languages and maintain a low complexity of reasoning [26]. In the previous study [27], the authors demonstrated that all the axioms asserted in DL-Lite$_{core}$ could be reduced to the ones with *subclassOf* and *disjointWith* relations. Therefore, our proposed model is mainly designed to learn the representations of axioms that are defined by these two relations. In order to better preserve the properties of relations and to measure the confidence of axioms in schematic knowledge, we implement CosE by projecting concepts into the angle-based semantic space and translation-based semantic space according to the type of relations. In the angle-based semantic space, each concept is encoded with one vector and a valid length, which are utilized to preserve the properties of these two kinds of relations. In the translation-based semantic space, the vector representations of concepts are employed to measure the confidence of related axioms. Furthermore, we design a negative sampling strategy according to the mutual exclusion relationship between *subclassOf* and *disjointWith* during the training process of CosE, which can learn better vector representations of concepts for completing schematic knowledge.

The main contributions of this work can be summarized as follows.

- We propose a cosine-based embedding model for completing lightweight schematic knowledge expressed in DL-Lite$_{core}$, in which two score functions are defined based on angle-based semantic space and translation-based semantic space so that the transitivity and symmetry of *subclassOf* and *disjointWith* relations can be preserved in our model simultaneously.
- We design a negative sampling strategy based on the mutual exclusion relationship between *subclassOf* axioms and *disjointWith* ones so that CosE can obtain better vector representations of concepts for schematic knowledge completion.
- We implement and evaluate our method based on four standard datasets constructed using real ontologies. Experiments on link prediction indicate that CosE could simultaneously preserve the logical properties (i.e., transitivity, symmetry) of relations and obtain better results than state-of-the-art models in most cases.

The rest of the paper is organized as follows. Section 2 shows the related work of knowledge graph embedding. Section 3 introduces the preliminaries, including both DL-Lite$_{core}$ and its logical properties. Our proposed model for completing the schematic knowledge is described in Section 4. Section 5 presents the experiments and evaluation results, followed by discussions in Section 6. Section 7 gives the conclusion and future directions of research.

## 2. Related Work

In this section, we briefly give an overview of the existing research efforts on KG embedding and divide them into two categories.

### 2.1. Factual Knowledge Embedding

Factual knowledge embedding mainly consists of two mainstream models, which are translational distance models and semantic matching models [9]. The former utilized the distance-based scoring function to measure the plausibility of one triple, and the latter used the similarity-based function to match the latent semantics of entities and relations in the vector space.

TransE [13] was one of the most representative translational distance models. It tried to encode both entities and relations of triples as vectors into the same semantic space. For each triple (h, r, t), the head entity and the tail entity in the semantic space were denoted by **h** and **t** that could be connected by their relations **t** with low error, such that $\mathbf{h} + \mathbf{r} \approx \mathbf{t}$. Afterward, several methods have been proposed to improve this idea. TransH [14] projected all the entities into one relation-specific hyper-plane, which made different roles of one entity in different relations. TransR [15] and TransD [28] still followed the strategy of TransH. They projected entities into relation-specific spaces using a projection matrix so that more complex relations (i.e., 1-to-N, N-to-1 and N-to-N) could be encoded. To seek the most reliable relation among two entities, TransA [29] introduced an adaptive Mahalanobis distance to define the score function so that it could handle complex relations more flexibly. Nevertheless, the translation-based embedding strategy only considers the local information of triples, which cannot make full use of the global information in KGs.

Another type of KG embedding models based on semantic matching adopted the architectures of neural network, which obtained encouraging results of KG completion, including MLP [30], NAM [31], and R-GCN [32]. In addition, ProjE [33] and ConvE [34] introduced the features of complex space and further optimized the score functions of underlying models. Therefore, both of them could obtain better performances compared to several models without the features of complex space.

The methods for factual knowledge embedding mainly consider encoding the triples of KGs to obtain vector representations of entities and relations for factual knowledge completion. However, few of them pay attention to model logical properties (e.g., transitivity, symmetry) of relations. Hence, these methods can hardly be applied to the related tasks of schematic knowledge or ontologies (e.g., completion, reasoning, and repairing).

### 2.2. Schematic Knowledge Embedding

The studies of schematic knowledge embedding are primarily composed of logical rules embedding, logical properties embedding, and ontology embedding [35].

As the RDF(S) and schematic knowledge can be transformed into logical rules, several studies have tried to encode them into embedding models and enhance the performances for knowledge completion [36]. Guo et al. [37] designed a joint model named KALE, which could simultaneously encode the triples of factual knowledge and their related logical rules. Furthermore, the authors proposed an improved model called RUGE [38] that could integrate the labeled triples, soft rules, and unlabeled triples into an iterative framework for learning their vector representations in the semantic space. Similarly, Zhang et al. [39] proposed a model named IterE. Different from above methods that mainly proposed to learn rules, the authors devoted to learning the embeddings of entities and rules at the same time, making full of their advantages complementing each other during the processing of model learning.

To further maintain the logical properties of relations, some works have been proposed for schematic knowledge and lightweight ontologies called RDF Schema. On2Vec [40] was a translation-based method for embedding the population of ontologies, in which the matrices were introduced to encode the transitivity of several relations. In order to encode concepts and instances into the same semantic space, EmbedS [20] and TransC [21] tried to encode concepts as spheres and instances as vectors so that the transitivity of the *is-A* relations could be preserved. To model the semantic hierarchies of KGs, Zhang et al. [22] proposed a method called HAKE for modeling semantic hierarchies. It is inspired by the fact that concentric circles can naturally reflect the hierarchy in the polar coordinate system. To further model composition among relations, RotatE [23] encoded entities and relations into a complex vector space, in which each relation was treated as a rotation from its related head entity to the tail entity. Furthermore, it could persevere the (anti)-symmetry and inversion of relations at the same time.

Recently, embedding models for ontologies have received attention. EL Embedding [24] and Quantum Embedding [41] were two representative algorithms based on the end-to-

end paradigm, in which loss functions and score functions were designed tailored for logical axioms expressed by $\mathcal{EL}^{++}$ and $\mathcal{ALC}$, respectively. These two embedding models encoded the semantics of the logical constructors by transforming the relations into geometric relations so that they could complete some kinds of axioms in ontologies very well. Chen et al. [25] proposed an ontology embedding model combined with word embedding and random walk algorithm, called OWL2Vec$^\star$, which took in the lexical information, logical constructors, and graph structures of ontologies so that it could preserve the semantics of ontologies well.

Although the above models are enabled to encode the logical properties of relations in their designed semantic spaces, it is still a challenge for them to preserve the transitivity (e.g., *subclassOf*) and symmetry (e.g., *disjointWith*) of axioms at the same time. To the best of our knowledge, our model is the first work for completing lightweight schematic knowledge expressed in DL-Lite$_{core}$, by which the transitivity and symmetry of relations in axioms can be simultaneously preserved well.

## 3. Preliminary

This section first gives the basic syntax and definition of DL-Lite$_{core}$. Then, we introduce the definition of schematic knowledge embedding, and formulate its properties for preserving the transitivity and symmetry of relations in DL-Lite$_{core}$.

### 3.1. DL-Lite$_{core}$

DL-Lite$_{core}$ is the core language for DL-Lite [26]. It is the lightweight language of Description Logics that represents the domain of interest via concepts denoting the set of instances and binary relations between instances. For the syntax of DL-Lite$_{core}$, the concepts and relations are defined as follows:

$$(1) \ B ::= A|\exists Q, \qquad\qquad (2) \ Q ::= P|P^-,$$
$$(3) \ C ::= B|\neg B, \qquad\qquad (4) \ R ::= Q|\neg Q,$$

where the symbols $A$ and $B$ denote an atomic concept and a basic one, $P$ and $Q$ represent an atomic relation and a basic one, and $C$ and $R$ denote the general concept and role, respectively.

The forms of axiom in DL-Lite$_{core}$ can be asserted as follows: (1) the inclusion axiom of concepts is denoted by $B \sqsubseteq C$; (2) the membership axiom of concept is denoted by $A(a)$, where $a$ is an individual; and (3) the membership axiom of relation is denoted by $P(a, b)$, where $a, b$ are two instances.

**Definition 1** ([42] (Ontology)). *Let $\mathcal{L}$ be a logical language from Description Logics. An ontology denoted by $O = \langle \mathcal{T}, \mathcal{A} \rangle$ consists of a TBox $\mathcal{T}$ and an ABox $\mathcal{A}$, where $\mathcal{T}$ is a set of concept inclusion axioms that is also called schematic knowledge, and $\mathcal{A}$ is a set of membership axioms about concepts and roles. The forms of all the axioms in O are constrained by the syntax of $\mathcal{L}$.*

For the lightweight schematic knowledge of KG, its logical axioms are mainly asserted based on *subclassOf* axioms and *disjointWith* axioms. According to the previous study [27], the authors demonstrated that all the axioms from TBox in DL-Lite$_{core}$ could be completely reduced to axioms with *subclassOf* and *disjointWith* relations using the transformational rules based on a directed graph. Hence, our proposed model can encode all the axioms from TBox expressed in DL-Lite$_{core}$.

Unless specified otherwise, we assume that all the axioms of lightweight schematic knowledge in the subsequent sections are expressed in DL-Lite$_{core}$, and we do not differentiate the TBox in DL-Lite$_{core}$ ontology and lightweight schematic knowledge. For convenience, we divide the axioms in TBox of DL-Lite$_{core}$ into two sets, denoted by $\mathcal{T} = \{(C_i, subclassOf, C_j)\} \cup \{(C_i, disjointWith, C_j)\}$, where $C_i$ and $C_j$ are two general concepts. Notice that the axioms are also one special kind of triples. For schematic knowl-

edge embedding, the encoded objects of models are head concepts, and the tail ones are defined by the syntax of DL-Lite$_{core}$, rather than entities without semantics.

Next, we give the definition of schematic knowledge embedding and formulate its properties for preserving the transitivity and symmetry of relations in DL-Lite$_{core}$.

*3.2. Schematic Knowledge Embedding for DL-Lite$_{core}$*

**Definition 2** (Schematic Knowledge Embedding). *Given a set of axioms* $\mathcal{T} = \{(C_i, r, C_j)\}$, *where $C_i, C_j$ are two concepts and $r$ is a relation of them. Its embedding model is a function $f(C_i, r, C_j) \rightarrow \mathbb{R}$ that can encode all the concepts and relations in $\mathcal{T}$ into the semantic space, in which the logical property of each relation $r$ can be preserved w.r.t. the asserted axioms $\{(C_i, r, C_j)\}$ and inferred ones at the numerical level.*

According to Definition 2, we formulate its properties for preserving the transitivity and symmetry of relations *subclassOf* and *disjointWith* in axioms during this process.

**Definition 3** (Schematic Knowledge Embedding for DL-Lite$_{core}$). *Given a TBox $\mathcal{T} = \{(C_i, r, C_j)\}$ expressed in DL-Lite$_{core}$, its embedding model denoted by $f(C_i, r, C_j)$ should satisfy the following properties to preserve the transitivity of subclassOf and symmetry of disjointWith:*

1.  *If $r = subclassOf$ and $(C_1, r, C_2)$, $(C_2, r, C_3)$ are two axioms asserted in $\mathcal{T}$, then $f(C_1, r, C_3) \approx f(C_1, r, C_2) \approx f(C_2, r, C_3)$.*
2.  *If $r = disjointWith$ and $(C_1, r, C_2)$ is an axiom asserted in $\mathcal{T}$, then $f(C_1, r, C_2) \approx f(C_2, r, C_1)$.*

Notice that the properties in Definition 3 are preconditions for the model of lightweight schematic knowledge embedding. The object function still needs to be competent for the task of completing lightweight schematic knowledge in DL-Lite$_{core}$. Our proposed model is subsequently designed to achieve these goals.

## 4. Method

In this section, we first show the framework of CosE for embedding schematic knowledge, and then we describe the score functions of CosE in detail. Finally, the strategy of negative sampling is introduced for training CosE.

*4.1. The Framework of CosE*

For each axiom $(C_i, r, C_j)$ with transitivity or symmetry relation $r$ expressed in DL-Lite$_{core}$, the existing KG embedding models prefer to treat relation $r$ as one single symbol, but they usually ignore its logical properties. Therefore, the transitivity and symmetry of relations cannot be preserved in the semantic space, which is hardly applied to schematic knowledge completion. To better complete lightweight schematic knowledge, we propose a cosine-based embedding model called CosE, which can simultaneously preserve the transitivity of *subclassOf* and symmetry of *disjointWith* well.

Figure 2 shows the framework of CosE for schematic knowledge embedding with a concrete example, where the relations *subclassOf* and *disjointWith* are denoted by solid lines and dotted lines, respectively. Given a set of axioms expressed in DL-Lite$_{core}$ shown in Figure 2a, CosE divides them into two disjoint sets, $S \cup D$ shown in Figure 2b, where $S$ and $D$ contain all the *subclassOf* axioms and *disjointWith* axioms, respectively. Then, all the concepts in $S$ and $D$ are projected according to the type of relations into angle-based semantic space and translation-based semantic space shown in Figure 2c. The angle-based semantic space is employed to preserve the transitivity or symmetry of relations in axioms, and the translation-based semantic space is used to measure the confidence of each axiom. Finally, the embedding of concepts will be obtained when the process of training CosE is finished.

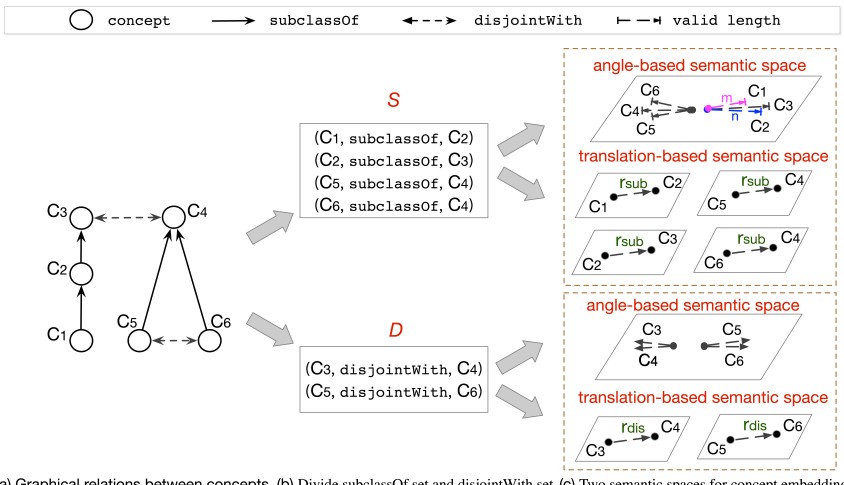

(a) Graphical relations between concepts  (b) Divide subclassOf set and disjointWith set  (c) Two semantic spaces for concept embedding

**Figure 2.** The framework of CosE for schematic knowledge embedding.

Notice that *subclassOf* and *disjointWith* relations are 1-to-N and N-to-N ones. For example, one concept could belong to (or disjoint with) several concepts. To measure the confidences of axioms more accurately, CosE introduces the mapping matrix $\mathbf{M}_{\mathbf{C_i C_j}}$ to encode concepts as vectors in translation-based semantic space, where $C_i$, $C_j$ are the concepts in given axioms. For the axiom $(C_1, subclassOf, C_2)$ shown in Figure 2c, the concepts $C_1$, $C_2$ will be projected by $\mathbf{M}_{\mathbf{C_i C_j}}$. It indicates that each axiom will be projected into one translation-based semantic space tailored for itself. More specifically, as shown in Figure 2c, we assume that $\mathbf{C_{1\perp}^{12}}$, $\mathbf{C_{2\perp}^{12}}$ are the projected vectors of $C_1$ and $C_2$ by $\mathbf{M}_{\mathbf{C_1 C_2}}$, and $\mathbf{C_{2\perp}^{23}}$ and $\mathbf{C_{3\perp}^{23}}$ are the projected vectors of $C_2$ and $C_3$ by $\mathbf{M}_{\mathbf{C_2 C_3}}$. It is easy to observe that the located translation-based semantic spaces of them are different, which is helpful for each axiom to obtain a suitable confidence in its projected space. Given one axiom $(C_i, r, C_j)$, we define its mapping matrix $\mathbf{M}_{\mathbf{C_i C_j}}$ as follows:

$$\mathbf{M}_{\mathbf{C_i C_j}} = \mathbf{C_{ip} C_{jp}^{\top}} + \mathbf{I^{n \times n}}, \tag{1}$$

where $\mathbf{C_{ip}}, \mathbf{C_{jp}} \in \mathbb{R}^n$ are the projection vectors of head concept $C_i$ and tail concept $C_j$ in axiom $(C_i, r, C_j)$. $\mathbf{I^{n \times n}}$ is an identity matrix. According to the defined mapping matrix $\mathbf{M}_{\mathbf{C_i C_j}}$, the projected vectors of concepts $C_i$ and $C_j$ in its translation-based semantic space are calculated as follows:

$$\mathbf{C_{i\perp}} = \mathbf{M}_{\mathbf{C_i C_j}} \mathbf{C_i}, \quad \mathbf{C_{j\perp}} = \mathbf{M}_{\mathbf{C_i C_j}} \mathbf{C_j}. \tag{2}$$

As shown in Figure 2c, the translation-based semantic space introduced in CosE can only measure the confidence of axioms, while the logical properties of relations need to be encoded through the angle-based semantic space. To deal with the transitivity of axioms $(C_1, subclassOf, C_2)$ and $(C_2, subclassOf, C_3)$ in $\mathcal{S}$, we hope the angles among vectors of $C_1$, $C_2$ and $C_3$ should be close to $0°$. To describe the direction of transmission about *subclassOf* axioms, CosE employs the vector length of each concept as a restriction in angle-based semantic space, in which the lengths of sub-concepts should be less than the ones of their parents. For the axioms $(C_1, subclassOf, C_2)$ and $(C_2, subclassOf, C_3)$ in $\mathcal{S}$, the length of $C_3$ is larger than $C_2$, whose length is larger than $C_1$. To preserve the symmetry, the length restrictions will be removed because the cosine function itself has the property of symmetry. For $(C_3, disjointWith, C_4)$, the vector representations of concepts $C_3$ and $C_4$ are similar in the angle-based semantic space. With the help of these two types of semantic spaces, the transitivity and symmetry of relations could be simultaneously preserved in CosE.

### 4.2. The Score Functions of CosE

As CosE projects the concepts of all the axioms into two types of semantic spaces, we define the corresponding score functions to evaluate the score of axioms in these two semantic spaces. Given an axiom $(C_i, r, C_j)$, its score function is defined as:

$$f(C_i, r, C_j) = f_a(C_i, r, C_j) + f_t(C_i, r, C_j), \tag{3}$$

where $f_a(C_i, r, C_j)$ is score function defined in the angle-based semantic space, and $f_t(C_i, r, C_j)$ is one designed for translation-based semantic space.

For the score function $f_a(C_i, r, C_j)$ in angle-based semantic space, we assume that the relations with different properties should be measured by different score functions. For an axiom $(C_i, r_s, C_j)$ with a *subclassOf* relation denoted by $r_s$, CosE encodes concepts $C_i$ and $C_j$ as $(\mathbf{C_i}, \mathbf{m})$ and $(\mathbf{C_j}, \mathbf{n})$, where $\mathbf{C_i}$ and $\mathbf{C_j}$ are the vectors of $C_i$ and $C_j$, and $\mathbf{m}$ and $\mathbf{n}$ are two external vectors introduced to obtain the valid lengths of $C_i$ and $C_j$ for preserving the direction of transmission between the projected concepts. The score function of axiom $f_a(C_i, r_s, C_j)$ designed for *subclassOf* relation is defined as follows.

$$f_a(C_i, r_s, C_j) = 1 - cos(\mathbf{C_i}, \mathbf{C_j}) + ||\mathbf{m}||_2 - ||\mathbf{n}||_2, \tag{4}$$

where $\mathbf{C_i} \in \mathbb{R}^n$ and $\mathbf{C_j} \in \mathbb{R}^n$ are two vectors corresponding to $C_i$ and $C_j$, and $||\mathbf{m}||_2$ and $||\mathbf{n}||_2$ are two valid lengths of $C_i$ and $C_j$. Note that these vectors are all parameters that need to be learned during the process of model training.

For an axiom $(C_i, r_d, C_j)$ with *disjointWith* relation denoted by $r_d$, the length constraints of vectors are removed so as to preserve the symmetry of the *disjointWith* relation. The score function corresponding to $(C_i, r_d, C_j)$ is defined as follows.

$$f_a(C_i, r_d, C_j) = 1 - cos(\mathbf{C_i}, \mathbf{C_j}), \tag{5}$$

where $\mathbf{C_i}, \mathbf{C_j} \in \mathbb{R}^n$ are the vectors of $C_i$ and $C_j$ in the angle-based semantic space.

Although the above score functions designed for angle-based semantic space can preserve the properties of *subclassOf* and *disjointWith*, it still cannot measure the confidences of axioms with these two relations well. This is because *subclassOf* and *disjointWith* are typical multivariate relations. To address this problem, we introduce a new score function for each axiom in translation-based semantic space to achieve this goal, as follows.

$$f_t(C_i, r, C_j) = ||\mathbf{C_{i\perp}} + \mathbf{r} - \mathbf{C_{j\perp}}||_2, \tag{6}$$

where $\mathbf{r}$ is the vector representation of relation $r$, and $\mathbf{C_{i\perp}}$ and $\mathbf{C_{j\perp}} \in \mathbb{R}^n$ are two projected vectors generated by Formulas (1) and (2) in translation-based semantic space. During the training process of CosE, we need to enforce the constraints such that $||\mathbf{C_i}||_2 \leq 1$, $||\mathbf{C_j}||_2 \leq 1$, $||\mathbf{C_{i\perp}}||_2 \leq 1$, and $||\mathbf{C_{j\perp}}||_2 \leq 1$.

Notice that the relation's own transitivity or symmetry can also be modeled via the function of angle-based semantic space, and the normal relations without logical properties can be modeled via the function of translation-based semantic space.

### 4.3. Negative Sampling Based on Schematic Knowledge for the Training Model

To train our proposed model, every axiom in the training set needs to be labeled as "positive" or "negative". However, there are only positive axioms asserted in the existing DL-Lite$_{core}$ ontologies. Thus, we need to corrupt the positive axioms to generate a set of negative axioms. Precisely, for each axiom $(C_i, r, C_j)$ asserted in the DL-Lite$_{core}$ ontologies, we adopt the negative sampling strategy to generate a set of negative axioms, by which $C_i$ or $C_j$ in $(C_i, r, C_j)$ are replaced with $(C_i', r, C_j)$ or $(C_i, r, C_j')$ according to the uniform probability distribution.

For all the axioms, we utilize the loss function based on margin rank to train the vector representation of concepts in CosE, where symbols $\mathcal{T}$ and $\mathcal{T}'$ denote two sets of positive

axioms and negative ones w.r.t. the type of relation. $\xi$ and $\xi'$ represent a positive axiom and a negative one selected from $\mathcal{T}$ and $\mathcal{T}'$, respectively. For the axioms with *subclassOf* relations, their loss function is defined as:

$$\mathcal{L}_{sub} = \sum_{\xi \in \mathcal{T}_{sub}} \sum_{\xi' \in \mathcal{T}'_{sub}} [\gamma_{sub} + f(\xi) - f(\xi')]_+, \tag{7}$$

where $f(\cdot)$ is the score function defined in Formula (3), and $\gamma_{sub}$ is a margin to separate the positive axiom and the negative one, $[x]_+ \stackrel{\triangle}{=} max(x, 0)$. Similarly, the loss function of axioms with *disjointWith* relations is defined as:

$$\mathcal{L}_{dis} = \sum_{\xi \in \mathcal{T}_{dis}} \sum_{\xi' \in \mathcal{T}'_{dis}} [\gamma_{dis} + f(\xi) - f(\xi')]_+. \tag{8}$$

Finally, the complete loss function of CosE is linearly composed of the above two loss functions, defined as follows.

$$\mathcal{L} = \mathcal{L}_{sub} + \mathcal{L}_{dis} \tag{9}$$

The target of training CosE is to minimize its loss function by updating the embeddings of concepts iteratively. Algorithm 1 presents the concrete procedure for training CosE. With a set of axioms $\mathcal{T}$ expressed in DL-Lite$_{core}$ as inputs, we first divide $\mathcal{T}$ into two disjoint sets, $\mathcal{S}$ and $\mathcal{D}$ (Lines 1–2). Line 3 initializes all the parameters related to concepts trained in CosE, denoted by $\mathcal{M}$. Lines 4–15 present the concrete realization of CosE for schematic knowledge embedding. For each axiom $(C_i, subclassOf, C_j)$ (or $(C_i, disjointWith, C_j)$), we employ the corresponding score function to learn the vector representations of concepts in two semantic spaces, and calculate the loss function of them (Lines 5–14). Line 15 calculates the sum of two loss functions of $\mathcal{L}_{sub}$ and $\mathcal{L}_{dis}$. The whole training process will be terminated until the loss function $\mathcal{L}$ of model $\mathcal{M}$ is converged.

---

**Algorithm 1:** The algorithm of training CosE model

---

　　**Input:** A set of axioms $\mathcal{T}$ expressed in DL-Lite$_{core}$.
　　**Output:** The trainded model with concepts embedding $\mathcal{M}$
1　$\mathcal{S} \leftarrow \{(C_i, subclassOf, C_j) | \forall (C_i, subclassOf, C_j) \in \mathcal{T}\}$;
　　// Obtain the set of *subclassOf* axioms
2　$\mathcal{D} \leftarrow \{(C_i, disjointWith, C_j) | \forall (C_i, disjointWith, C_j) \in \mathcal{T}\}$;
　　// Obtain the set of *disjointWith* axioms
3　Initial all the parameters $\mathbf{C_i}$, $\mathbf{C_j}$, $\mathbf{C_{i\perp 2}}$, $\mathbf{C_{j\perp}}$, $\mathbf{m}$ and $\mathbf{n}$ related to concepts in $\mathcal{M}$;
4　**while** *the loss function $\mathcal{L}$ of model $\mathcal{M}$ is not converged* **do**
5　　**for** *each axiom $(C_i, subclassOf, C_j) \in S$* **do**
6　　　Learn the vector representations of $C_i, C_j$ by the score function $f_a(C_i, r_s, C_j)$;
7　　　Learn the vector representations of $C_i, C_j$ by the score function $f_t(C_i, r, C_j)$;
8　　**end**
9　　Calculate the loss function $\mathcal{L}_{sub}$ according to Formula (7);
10　　**for** *each axiom $(C_i, disjointWith, C_j) \in D$* **do**
11　　　Learn the vector representations of $C_i, C_j$ by the score function $f_a(C_i, r_d, C_j)$;
12　　　Learn the vector representations of $C_i, C_j$ by the score function $f_t(C_i, r, C_j)$;
13　　**end**
14　　Calculate the loss function $\mathcal{L}_{dis}$ according to Formula (8);
15　　$\mathcal{L} \leftarrow \mathcal{L}_{sub} + \mathcal{L}_{dis}$;
16　**end**
17　**return** $\mathcal{M}$;

---

Furthermore, we design a novel negative sampling skill according to the mutual exclusion between *subclassOf* and *disjointWith* relations. Unlike the uniform negative sampling method that randomly samples its replacer from all the concepts, we restrict the sampling

scope to a group of candidates, which can provide more meaningful information during the process of training. Precisely, for each axiom $(C_i, r_s, C_j)$ with $subclassOf$ relation $r_s$, if there exist $subclassOf$ relations (e.g., $(C_i', subclassOf, C_i)$ or $(C_j, subclassOf, C_j')$) asserted or inferred in ontologies, we need to exclude these replace cases because of the transitivity of $subclassOf$. Relatively, if there exist $disjointWith$ relations (e.g., $(C_i', disjointWith, C_i)$ or $(C_j, disjointWith, C_j')$) in ontologies, we need to give the highest priority to these relations for replace cases. Similarly, for each axiom $(C_i, r_d, C_j)$ with $disjointWith$ relation $r_d$, we need to exclude the replace cases $(C_i', subclassOf, C_i)$ or $(C_j', subclassOf, C_j)$, which are asserted or inferred in ontologies. With these semantic constraints for negative sampling, we can obtain better vector representations of concepts for completing schematic knowledge.

## 5. Experiments and Results

To evaluate our method, we compare CosE with several well-known models and state-of-the-art methods of KG embedding on link prediction, which is a typical task employed for knowledge graph completion. In addition, we further extend the tasks of link prediction in order to verify the ability of models for preserving the transitivity and symmetry of relations in schematic knowledge.

### 5.1. Datasets

Although there exist several benchmark datasets (e.g., FB15K, WN18) in previous works [13–15], it is not suitable for them to evaluate the models for completing schematic knowledge. This is because most datasets mainly consist of factual knowledge, but few of them contain enough concepts and related axioms. Therefore, we collect four lightweight schematic knowledge named YAGO-On, FMA [43], FoodOn [44], and HeLiS [45], and two variants built based on YAGO-On (i.e., YAGO-on-t and YAGO-on-s), listed as follows.

- **YAGO-On**: It is built from the well-known knowledge graph YAGO [4], which contains lots of concepts from WordNet [2].
- **FMA**: It is an evolving ontology that has been maintained by University of Washington since 1994. It conceptualizes the phenotype structure of human in a machine-readable form, whose biomedical schematic knowledge has been open source in OAEI (http://oaei.ontologymatching.org/ accessed on 5 September 2022).
- **FoodOn**: It is a comprehensive ontology resource that spans various domains related to food, which can precisely describe foods commonly known in cultures from around the world.
- **HeLiS**: It is an ontology for promoting healthy lifestyles, which tries to conceptualize the domains of food and physical activity so that some unhealthy behaviors can be monitored.
- **YAGO-On-t**: It is built from the axioms in YAGO-On according to the transitivity property of $subclassOf$. If there exist $(C_i, subclassOf, C_j)$ and $(C_j, subclassOf, C_m)$ in YAGO-On, we add an axiom $(C_i, subclassOf, C_m)$ to YAGO-On-t.
- **YAGO-On-s**: It is built from the axioms in YAGO-On according to the symmetry property of $disjointWith$. If there exists an axiom $(C_i, disjointWith, C_j)$ in YAGO-On, we add an axiom $(C_j, disjointWith, C_i)$ to YAGO-On-s.

As several datasets only contain axioms with $subclassOf$ relations, we need to supplement some $disjointWith$ axioms. To achieve this goal, we utilize heuristic rules proposed in [46] to generate some axioms with $disjointWith$ relations and inject them into the original datasets. Table 1 lists the statistics of the above datasets for evaluation.

**Table 1.** The statistics of generated datasets for evaluation.

| Dataset | | YAGO-On [4] | FMA [43] | FoodOn [44] | HeLiS [45] | YAGO-On-t | YAGO-On-s |
|---|---|---|---|---|---|---|---|
| ♯ Concept | | 46,109 | 78,988 | 28,182 | 17,550 | 46,109 | 46,109 |
| Train | ♯ *subclassOf* | 29,181 | 29,181 | 20,844 | 14,222 | 11,898 | 0 |
| | ♯ *disjointWith* | 32,673 | 32,673 | 17,398 | 13,782 | 0 | 10,000 |
| Valid | ♯ *subclassOf* | 1000 | 2000 | 1488 | 1015 | 1000 | 1000 |
| | ♯ *disjointWith* | 1000 | 2000 | 2714 | 1722 | 1000 | 1000 |
| Test | ♯ *subclassOf* | 1000 | 2000 | 2978 | 2032 | 5949 | 0 |
| | ♯ *disjointWith* | 1000 | 1000 | 2174 | 1722 | 0 | 10,000 |

♯: indicates number of concepts.

*5.2. Implementation Details*

To verify the effectiveness of CosE, we employ several KG embedding models as baselines, including TransE [13], TransH [14], TransR [15], TransD [28], RESCAL [16], HolE [18], ComplEx [19], and Analogy [47], which are implemented by the OpenKE platform [48]. In addition, we utilize the state-of-art KG embedding methods (i.e., TransC [21], RotatE [23], and EL Embedding [24]) (For the fairness of comparison, we do not compare CosE with OWL2Vec* [25] because it makes full of labels, comments, and extra resources of ontologies, whereas its source codes cannot be split.) to compare with CosE.

We implement CosE in Python with the help of the PyTorch platform. Its source code can be downloaded along with datasets (https://github.com/zhengxianda/CosE accessed on 5 September 2022). We utilize the stochastic gradient descent (SGD) with the mini-batch strategy to train CosE, and employ SGD as an optimizer to fine-tune hyper-parameters according to the validation datasets. The ranges of several hyper-parameters are listed as follows: the dimension $d$ for embedding concepts is selected from the scope of $\{100, 125, 200, 250, 500, 1000\}$, the mini-batch size $b$ for the training range of $\{64, 128, 200, 512, 1024, 2048\}$, and the margin $\gamma$ for the loss functions range of $\{1, 2, 3, 6, 9, 12, 15\}$. For some special models (e.g., ComplEx, RotatE), we adopt uniform initialization for the real and imaginary vectors of concepts and relations. Notice that we do not employ regularization to constrain CosE because we observe that the fixed margin $\gamma$ can effectively prevent CosE from over-fitting. The best configurations of hyper-parameters are determined by the validation set in terms of mean rank. Finally, the optimized hyper-parameters of CosE are $d = 200$, $b = 200$, and $\gamma = 3$. In order to distinguish the effect between our proposed strategy of negative sampling and the traditional one, the symbol "CosE" represents that our negative sampling strategy has been equipped as the default, and "CosE"$^-$ adopts the traditional negative sampling strategy in the subsequent tables.

*5.3. The Results of Link Prediction*

Link prediction is a typical task for completing the axiom when one of the concepts or their relation is missing. We employ *MRR* and *Hits@N* as evaluation metrics proposed in TransE [13]. For each axiom $(C_i, r, C_j)$ in test datasets, we replace concept $C_i$ or $C_j$ with $C_n$ in the set of concepts $\mathcal{C}$ to generate *corrupted axioms*, and measure the confidences of these axioms using the score function. Then, the rank of correct concepts could be obtained by sorting the confidences of axioms in descending order. *MRR* is a metric that calculates the mean reciprocal rank of all correct concepts. *Hits@N* is a metric that counts the ratio of the correct concepts ranked in the *top N*. Notice that the corrupted axiom ranking above a test axiom is also valid, and should not be treated as a wrong axiom. Therefore, corrupted axioms asserted in schematic knowledge have been filtered before ranking. For convenience, we label the filtered result as "Filter", and the unfiltered one is denoted by "Raw". In our experiments of link prediction, all the models are required to infer the missing concept $C_i$ or $C_j$ for the axioms $(C_i, r, C_j)$ in the test datasets. For the metrics *MRR* and *Hits@N*, a higher value of them indicates a better performance of the evaluated model.

Tables 2 and 3 list the results of link prediction on YAGO-On, FMA, FoodOn, and HeLiS. Overall, CosE⁻ and CosE have obviously surpassed other models in terms of *MRR* and *Hits@N*. It indicates that both of them can preserve the logical properties (i.e., transitivity, symmetry) of relations via two designed semantic spaces, which can help our model to learn better vector representations of concepts for completing schematic knowledge. Compared with models employing the strategy of projection matrices (e.g., TransH, TransR and TransD), CosE is able to measure the confidences of related axioms more precisely. The possible reasons are that CosE attempts to project axioms with different relations into different translation-based semantic spaces, and the types of relations in schematic knowledge are relatively small. Hence, the projection strategy of CosE is more suitable than with other models. Furthermore, benefiting from our proposed strategy of negative sampling, the performances of CosE are slightly better than CosE⁻ in terms of *MRR* and *Hits@N*. We analyze that the mutual exclusion between *subclassOf* and *disjointWith* is useful in distinguishing the similar embedding of concepts in the semantic space.

**Table 2.** The results of YAGO-On and FMA on link prediction.

| Metric | YAGO-On | | | | | FMA | | | | |
|---|---|---|---|---|---|---|---|---|---|---|
| | MRR | | Hits@N(%) | | | MRR | | Hits@N(%) | | |
| | Raw | Filter | 10 | 3 | 1 | Raw | Filter | 10 | 3 | 1 |
| TransE [13] | 0.241 † | 0.501 † | 0.784 † | 0.582 † | 0.343 † | 0.066 † | 0.325 † | 0.474 † | 0.371 † | 0.247 † |
| TransH [14] | 0.195 † | 0.196 † | 0.472 † | 0.252 † | 0.091 † | 0.008 † | 0.009 † | 0.018 † | 0.005 † | 0.003 † |
| TransR [15] | 0.090 † | 0.428 † | 0.588 † | 0.433 † | 0.355 † | 0.060 † | 0.411 † | 0.490 † | 0.440 † | 0.370 † |
| TransD [28] | 0.038 † | 0.176 † | 0.462 † | 0.305 † | 0.000 † | 0.034 † | 0.149 † | 0.430 † | 0.250 † | 0.000 † |
| RESCAL [16] | 0.080 † | 0.339 † | 0.525 † | 0.392 † | 0.244 † | 0.047 † | 0.317 † | 0.469 † | 0.377 † | 0.236 † |
| HolE [18] | 0.155 | 0.231 | 0.523 | 0.254 | 0.099 | 0.039 | 0.112 | 0.311 | 0.120 | 0.033 |
| ComplEx [19] | 0.034 † | 0.237 † | 0.491 † | 0.403 † | 0.058 † | 0.033 † | 0.201 † | 0.484 † | 0.372 † | 0.011 † |
| Analogy [47] | 0.037 † | 0.301 † | 0.496 † | 0.429 † | 0.160 † | 0.037 † | 0.277 † | 0.487 † | 0.415 † | 0.130 † |
| TransC [21] | 0.112 * | 0.420 * | 0.698 * | 0.502 * | 0.298 * | – | – | – | – | – |
| RotatE [23] | 0.002 | 0.002 | 0.001 | 0.000 | 0.000 | 0.001 | 0.001 | 0.001 | 0.000 | 0.000 |
| EL Embedding [24] | 0.008 | 0.008 | 0.005 | 0.000 | 0.000 | 0.014 | 0.014 | 0.019 | 0.001 | 0.001 |
| CosE⁻ | 0.229 | 0.558 | 0.859 | 0.648 | 0.495 | 0.093 | 0.386 | 0.628 | 0.391 | 0.271 |
| CosE | **0.247** | **0.657** | **0.861** | **0.714** | **0.550** | **0.117** | **0.507** | **0.640** | **0.545** | **0.423** |

† Indicates that the results are taken from our published work [49]. Other results are obtained by their source codes. * As experimental results of TransC are much worse than the ones mentioned in the paper [21], we utilize its original results for evaluation.

**Table 3.** The results of FootOn and HeLiS on link prediction.

| Metric | FoodOn | | | | | HeLiS | | | | |
|---|---|---|---|---|---|---|---|---|---|---|
| | MRR | | Hits@N(%) | | | MRR | | Hits@N(%) | | |
| | Raw | Filter | 10 | 3 | 1 | Raw | Filter | 10 | 3 | 1 |
| TransE [13] | 0.011 | 0.012 | 0.020 | 0.011 | 0.006 | 0.037 | 0.037 | 0.078 | 0.028 | 0.010 |
| TransH [14] | 0.010 | 0.012 | 0.020 | 0.013 | 0.006 | 0.026 | 0.026 | 0.050 | 0.020 | 0.006 |
| TransR [15] | 0.008 | 0.008 | 0.013 | 0.008 | 0.004 | 0.025 | 0.026 | 0.056 | 0.016 | 0.004 |
| TransD [28] | 0.003 | 0.003 | 0.007 | 0.004 | 0.000 | 0.008 | 0.008 | 0.018 | 0.005 | 0.000 |
| RESCAL [16] | 0.001 | 0.001 | 0.004 | 0.000 | 0.000 | 0.003 | 0.003 | 0.004 | 0.003 | 0.001 |
| HolE [18] | 0.002 | 0.002 | 0.007 | 0.001 | 0.000 | 0.035 | 0.035 | 0.078 | 0.024 | 0.008 |
| ComplEx [19] | 0.001 | 0.001 | 0.003 | 0.000 | 0.000 | 0.001 | 0.001 | 0.001 | 0.000 | 0.000 |
| RotatE [23] | 0.009 | 0.009 | 0.017 | 0.008 | 0.004 | 0.023 | 0.023 | 0.033 | 0.026 | 0.012 |
| EL Embedding [24] | 0.001 | 0.001 | 0.002 | 0.001 | 0.000 | 0.001 | 0.001 | 0.001 | 0.000 | 0.000 |
| CosE⁻ | 0.032 | 0.037 | 0.080 | **0.058** | 0.009 | 0.077 | 0.077 | 0.144 | 0.079 | 0.034 |
| CosE | **0.034** | **0.038** | **0.083** | 0.057 | **0.011** | **0.080** | **0.080** | **0.152** | **0.081** | **0.035** |

Tables 4 and 5 list the results of link prediction about the *subclassOf* axioms of four datasets. Overall, CosE⁻ and CosE outperform most models in terms of *MRR* and *Hits@N*.

It shows that the transitivity of *subclassOf* is preserved well in our defined semantic spaces. Relatively, the performances of some schematic knowledge embedding methods are not well. We analyze that most of their score functions focus on modeling *subclassOf* relations via spheres, but the external *disjointWith* axioms may influence the convergence of their score functions.

**Table 4.** The results of YAGO-On and FMA on link prediction about *subclassOf* axioms.

| | YAGO-On | | | | | FMA | | | | |
|---|---|---|---|---|---|---|---|---|---|---|
| **Metric** | **MRR** | | **Hits@N(%)** | | | **MRR** | | **Hits@N(%)** | | |
| | **Raw** | **Filter** | **10** | **3** | **1** | **Raw** | **Filter** | **10** | **3** | **1** |
| TransE [13] | 0.375 [†] | 0.375 [†] | 0.722 [†] | **0.472** [†] | 0.179 [†] | 0.113 [†] | 0.113 [†] | 0.260 [†] | 0.110 [†] | 0.035 [†] |
| TransH [14] | 0.377 [†] | 0.377 [†] | 0.494 [†] | 0.179 [†] | 0.179 [†] | 0.110 [†] | 0.110 [†] | **0.295** [†] | 0.080 [†] | 0.040 [†] |
| TransR [15] | 0.063 [†] | 0.063 [†] | 0.216 [†] | 0.020 [†] | 0.000 [†] | 0.010 [†] | 0.010 [†] | 0.050 [†] | 0.050 [†] | 0.050 [†] |
| TransD [28] | 0.011 [†] | 0.011 [†] | 0.018 [†] | 0.008 [†] | 0.000 [†] | 0.050 [†] | 0.050 [†] | 0.050 [†] | 0.000 [†] | 0.000 [†] |
| RESCAL [16] | 0.069 [†] | 0.069 [†] | 0.143 [†] | 0.073 [†] | 0.035 [†] | 0.009 [†] | 0.009 [†] | 0.010 [†] | 0.005 [†] | 0.005 [†] |
| HolE [18] | 0.225 | 0.225 | 0.434 | 0.229 | 0.126 | 0.002 | 0.002 | 0.000 | 0.000 | 0.000 |
| ComplEx [19] | 0.001 [†] | 0.003 [†] | 0.002 [†] | 0.001 [†] | 0.001 [†] | 0.003 [†] | 0.003 [†] | 0.010 [†] | 0.000 [†] | 0.000 [†] |
| Analogy [47] | 0.003 [†] | 0.003 [†] | 0.035 [†] | 0.003 [†] | 0.003 [†] | 0.050 [†] | 0.050 [†] | 0.050 [†] | 0.050 [†] | 0.050 [†] |
| RotatE [23] | 0.001 | 0.001 | 0.000 | 0.000 | 0.000 | 0.002 | 0.002 | 0.002 | 0.000 | 0.000 |
| EL Embedding [24] | 0.001 | 0.001 | 0.000 | 0.000 | 0.000 | 0.001 | 0.001 | 0.000 | 0.000 | 0.000 |
| CosE⁻ | 0.393 | 0.393 | 0.724 | 0.471 | **0.226** | 0.128 | 0.128 | **0.295** | **0.165** | 0.030 |
| CosE | **0.397** | **0.397** | **0.726** | 0.458 | 0.240 | **0.145** | **0.145** | 0.290 | 0.140 | **0.065** |

† Shows that the results are taken from the work [49]. Other results are generated by their source codes.

**Table 5.** The results of FoodOn and HeLiS on link prediction about *subclassOf* axioms.

| | FoodOn | | | | | HeLiS | | | | |
|---|---|---|---|---|---|---|---|---|---|---|
| **Metric** | **MRR** | | **Hits@N(%)** | | | **MRR** | | **Hits@N(%)** | | |
| | **Raw** | **Filter** | **10** | **3** | **1** | **Raw** | **Filter** | **10** | **3** | **1** |
| TransE [13] | 0.015 | 0.015 | 0.024 | 0.014 | 0.009 | 0.072 | 0.072 | 0.154 | 0.052 | 0.019 |
| TransH [14] | 0.011 | 0.014 | 0.025 | 0.015 | 0.006 | 0.050 | 0.050 | 0.097 | 0.037 | 0.012 |
| TransR [15] | 0.011 | 0.011 | 0.017 | 0.010 | 0.006 | 0.050 | 0.050 | 0.111 | 0.032 | 0.008 |
| TransD [28], | 0.001 | 0.001 | 0.001 | 0.000 | 0.000 | 0.016 | 0.016 | 0.035 | 0.011 | 0.000 |
| RESCAL [16] | 0.001 | 0.001 | 0.001 | 0.000 | 0.000 | 0.005 | 0.005 | 0.007 | 0.005 | 0.003 |
| HolE [18] | 0.004 | 0.004 | 0.013 | 0.002 | 0.000 | 0.070 | 0.070 | 0.155 | 0.048 | 0.016 |
| ComplEx [19] | 0.001 | 0.001 | 0.000 | 0.000 | 0.000 | 0.001 | 0.001 | 0.000 | 0.000 | 0.000 |
| RotatE [23] | 0.017 | 0.017 | 0.033 | 0.016 | 0.017 | 0.046 | 0.046 | 0.065 | 0.051 | 0.024 |
| EL Embedding [24] | 0.001 | 0.001 | 0.001 | 0.001 | 0.000 | 0.001 | 0.001 | 0.000 | 0.000 | 0.000 |
| CosE⁻ | 0.038 | 0.038 | 0.074 | **0.041** | 0.017 | 0.152 | 0.152 | 0.286 | 0.155 | 0.068 |
| CosE | **0.040** | **0.040** | **0.079** | 0.036 | **0.021** | **0.158** | **0.158** | **0.300** | **0.159** | **0.071** |

Tables 6 and 7 list the results of the axioms with *disjointWith* relations. In terms of *MRR* and *Hits@N*, CosE⁻ and CosE have outperformed all the models in most cases. For link prediction results on YAGO-On and FMA, the performances of CosE are a little worse than TransR and TransE in *MRR Raw*, but it obtains better results in terms of *MRR Filter*. Through further analysis, we discover that CosE tends to give a higher score to the correct corrupted axiom, so the value of *MRR Raw* in CosE is much smaller than its *MRR Filter* value. Overall, *Hits@1* of CosE has been improved from 15% to 30% in YAGO-On and FMA. It shows that CosE could preserve the symmetry of relations precisely in the angle-based semantic space. Nevertheless, the performances of all the KG embedding models on FoodOn and HeLiS are unsatisfactory. We observe that it may be related to the generation method [46] of *disjointWith* axioms. More importantly, it indicates that the current models still lack scalability for various large-scale ontologies to some extent, and we will leave these issues for future work.

**Table 6.** The results of YAGO-On and FMA on link prediction about *disjointWith* axioms.

| Metric | YAGO-On | | | | | FMA | | | | |
|---|---|---|---|---|---|---|---|---|---|---|
| | MRR | | Hits@N(%) | | | MRR | | Hits@N(%) | | |
| | Raw | Filter | 10 | 3 | 1 | Raw | Filter | 10 | 3 | 1 |
| TransE [13] | 0.120 † | 0.627 † | 0.846 † | 0.693 † | 0.507 † | **0.122** † | 0.639 † | 0.927 † | 0.741 † | 0.491 † |
| TransH [14] | 0.010 † | 0.014 † | 0.220 † | 0.010 † | 0.003 † | 0.005 † | 0.006 † | 0.002 † | 0.001 † | 0.001 † |
| TransR [15] | **0.132** † | 0.792 † | 0.974 † | 0.848 † | 0.710 † | 0.010 † | 0.010 † | 0.050 † | 0.050 † | 0.050 † |
| TransD [15] | 0.066 † | 0.774 † | 0.906 † | 0.621 † | 0.000 † | 0.066 † | 0.292 † | 0.873 † | 0.488 † | 0.000 † |
| RESCAL [16] | 0.100 † | 0.640 † | 0.920 † | 0.720 † | 0.500 † | 0.094 † | 0.640 † | 0.940 † | 0.750 † | 0.480 † |
| HolE [18] | 0.084 | 0.237 | 0.611 | 0.278 | 0.072 | 0.078 | 0.224 | 0.622 | 0.240 | 0.065 |
| ComplEx [19] | 0.066 † | 0.470 † | 0.970 † | 0.820 † | 0.110 † | 0.003 † | 0.003 † | 0.010 † | 0.000 † | 0.000 † |
| Analogy [47] | 0.074 † | 0.598 † | 0.988 † | 0.854 † | 0.317 † | 0.069 † | 0.557 † | 0.979 † | 0.823 † | 0.264 † |
| RotatERotatE [23] | 0.001 | 0.001 | 0.000 | 0.000 | 0.000 | 0.001 | 0.001 | 0.000 | 0.000 | 0.000 |
| EL Embedding [24] | 0.016 | 0.016 | 0.010 | 0.001 | 0.000 | 0.028 | 0.028 | 0.037 | 0.002 | 0.001 |
| CosE⁻ | 0.066 | 0.723 | 0.994 | 0.824 | 0.764 | 0.058 | 0.644 | 0.962 | 0.617 | 0.512 |
| CosE | 0.097 | **0.917** | **0.996** | **0.970** | **0.860** | 0.090 | **0.870** | **0.990** | **0.950** | **0.780** |

† shows that the results are taken from the work [49]. Other results are obtained by their source codes.

**Table 7.** The results of FoodOn and HeLiS on link prediction about *disjointWith* axioms.

| Metric | FoodOn | | | | | HeLiS | | | | |
|---|---|---|---|---|---|---|---|---|---|---|
| | MRR | | Hits@N(%) | | | MRR | | Hits@N(%) | | |
| | Raw | Filter | 10 | 3 | 1 | Raw | Filter | 10 | 3 | 1 |
| TransE [13] | 0.007 | 0.007 | 0.015 | 0.007 | 0.004 | 0.001 | **0.002** | **0.003** | 0.002 | 0.000 |
| TransH [14] | 0.010 | 0.010 | 0.015 | 0.011 | 0.006 | **0.002** | **0.002** | **0.003** | 0.002 | 0.000 |
| TransR [15] | 0.005 | 0.005 | 0.010 | 0.005 | 0.002 | 0.001 | 0.001 | 0.002 | 0.001 | 0.000 |
| TransD [28] | 0.005 | 0.006 | 0.014 | 0.009 | 0.001 | 0.001 | 0.001 | 0.002 | 0.000 | 0.000 |
| RESCAL [16] | 0.001 | 0.001 | 0.001 | 0.000 | 0.000 | 0.001 | 0.001 | 0.001 | 0.001 | 0.000 |
| HolE [18] | 0.001 | 0.001 | 0.001 | 0.001 | 0.000 | 0.001 | 0.001 | 0.002 | 0.001 | 0.000 |
| ComplEx [19] | 0.001 | 0.001 | 0.001 | 0.000 | 0.000 | 0.001 | 0.001 | 0.001 | 0.000 | 0.000 |
| RotatE [23] | 0.001 | 0.001 | 0.001 | 0.001 | 0.000 | 0.001 | 0.001 | 0.001 | 0.001 | 0.000 |
| EL Embedding [24] | 0.001 | 0.001 | 0.003 | 0.001 | 0.000 | 0.001 | 0.001 | 0.001 | 0.001 | 0.000 |
| CosE⁻ | 0.023 | 0.035 | 0.085 | 0.075 | 0.000 | 0.001 | **0.002** | **0.003** | **0.003** | 0.000 |
| CosE | **0.027** | **0.036** | **0.087** | **0.078** | 0.000 | 0.001 | **0.002** | **0.003** | **0.003** | 0.000 |

*5.4. The Results of Transitivity and Symmetry*

As discussed above, we further verify whether the embeddings of concepts in CosE can encode the transitivity and symmetry implicitly. To achieve this goal, we design two experiments based on link prediction using the constructed datasets (i.e., YAGO-On-t, YAGO-On-s). In YAGO-On-t, if the related axioms $(C_i, subclassOf, C_j)$ and $(C_j, subclassOf, C_m)$ in the training set are satisfied by the transitivity rule, the testing set will contain the inferred axiom $(C_i, subclassOf, C_m)$ according to the transitivity property of *subclassOf*. Then, we train embedding models using the training set of YAGO-On-t, and utilize link prediction on the testing set to evaluate their performances on transitivity. Analogously, we evaluate the symmetry of models using YAGO-On-s. If the training set contains $(C_i, disjointWith, C_j)$, the axiom $(C_j, disjointWith, C_i)$ will be added to the testing set.

As listed in Table 8, we observe that CosE⁻ and CosE achieve better results than other models on the constructed datasets in most cases. On YAGO-On-t, the performances of CosE have exceeded other models in terms of *MRR* and *Hits@N*. On YAGO-On-s, CosE is slightly worse than HolE and TransE in terms of *MRR* and *Hits@1*. It shows that CosE has better potential than other models to perform schematic knowledge completion in terms of transitivity and symmetry.

**Table 8.** The evaluated results on link prediction of transitivity and symmetry.

| Metric | YAGO-On-t | | | | | YAGO-On-s | | | | |
|---|---|---|---|---|---|---|---|---|---|---|
| | MRR | | Hits@N(%) | | | MRR | | Hits@N(%) | | |
| | Raw | Filter | 10 | 3 | 1 | Raw | Filter | 10 | 3 | 1 |
| TransE [13] | 0.064 † | 0.077 † | 0.142 † | 0.070 † | 0.001 † | 0.043 † | **0.369** † | 0.971 † | 0.514 † | 0.080 † |
| TransH [14] | 0.200 † | 0.238 † | 0.309 † | 0.214 † | 0.149 † | 0.001 † | 0.002 † | 0.001 † | 0.000 † | 0.000 † |
| TransR [15] | 0.012 † | 0.013 † | 0.003 † | 0.002 † | 0.001 † | 0.010 † | 0.010 † | 0.001 † | 0.000 † | 0.000 † |
| TransD [28] | 0.008 † | 0.009 † | 0.020 † | 0.001 † | 0.000 † | 0.001 † | 0.181 † | 0.512 † | 0.302 † | 0.000 † |
| RESCAL [16] | 0.016 † | 0.020 † | 0.055 † | 0.015 † | 0.004 † | 0.032 † | 0.166 † | 0.449 † | 0.226 † | 0.039 † |
| HolE [18] | 0.040 | 0.045 | 0.082 | 0.008 | 0.002 | **0.070** | 0.342 | 0.716 | 0.425 | **0.128** |
| ComplEx [19] | 0.001 † | 0.001 † | 0.001 † | 0.000 † | 0.000 † | 0.036 † | 0.253 | 0.743 | 0.439 | 0.000 |
| Analogy [47] | 0.001 † | 0.001 † | 0.001 † | 0.001 † | 0.000 † | 0.043 † | 0.315 † | 0.932 † | 0.538 † | 0.000 † |
| RotatE [23] | 0.001 | 0.001 | 0.001 | 0.000 | 0.000 | 0.002 | 0.002 | 0.001 | 0.000 | 0.000 |
| EL Embedding [24] | 0.001 | 0.001 | 0.001 | 0.000 | 0.000 | 0.003 | 0.003 | 0.075 | 0.017 | 0.003 |
| CosE$^-$ | 0.203 | 0.260 | 0.403 | 0.246 | 0.177 | 0.038 | 0.322 | 0.983 | 0.554 | 0.000 |
| CosE | **0.207** | **0.284** | **0.408** | **0.261** | **0.218** | 0.038 | 0.323 | **0.992** | **0.557** | 0.000 |

† Shows that the results are taken from the work [49]. Other results are obtained via their source codes.

*5.5. Case Study*

The above experiments show that CosE has good performances for link prediction and can preserve the logical properties of transitivity or symmetry. Next, we present several concrete examples of completed results about CosE compared with TransE listed in Table 9, where the **bold** words show correct answers.

**Table 9.** The examples of predicted results on CosE compared with TransE.

| Head Concept | Relation | CosE | TransE [13] |
|---|---|---|---|
| Taksim_SK_footballers | subclassOf | person<br>player<br>site<br>club<br>**football_player** | person<br>airport<br>model<br>peninsula<br>singer |
| Soccer_clubs_in_the_<br>Greater_Los_Angeles_Area | subclassOf | site<br>person<br>player<br>**club**<br>football_player | person<br>airport<br>model<br>singer<br>writer |
| Tail Concept | Relation | CosE | TransE |
| Irish_male_models | DisjointWith | Filipino_female_models<br>**African_American_models**<br>LGBT_models | LGBT_models<br>South_African_female_models<br>American_male_models |

We observe that CosE can improve the performance of predicting the tail concepts compared with TransE. The first example shows the result of predicting the tail concept of one *subclassOf* axiom. For TransE, the correct answer is ranked 35th. The ranking of the correct answer in CosE has been significantly improved, and it is ranked fifth. It indicates that CosE could measure the confidences of axioms more precisely.

Table 9 also shows the ability of CosE to preserve transitivity and symmetry. Compared with TransE, the second example shows that CosE improves the correct answer of a *subclassOf* axiom from 24th to 4th. Similarly, the correct answer of the disjointness axiom in the third example is improved to second place. These examples indicate that CosE can infer the tail or head concept more precisely than existing KG embedding models for given axioms.

## 6. Discussion and Limitations

Compared to our preliminary work [49], we model the schematic knowledge embedding in view of lightweight ontology language called DL-Lite$_{core}$. Therefore, the inferred properties of DL-Lite$_{core}$ can be formulated and employed to optimize our methods. Notice that the mutual exclusion relationship between the *subclassOf* axioms and the *disjointWith* ones is derived from the minimal incoherence-preserving sub-TBoxes [50], which is the motivation of our negative sampling strategy. Therefore, our work will have a positive impact on optimizing schematic knowledge embedding models using the inferred properties of ontology language.

Secondly, we discover the result that the performances of all models for schematic knowledge completion on FoodOn and HeLiS are unsatisfactory. On the one hand, the reason is that existing models for completing schematic knowledge still lack the scalability for various large-scale ontologies to some extent. On the other hand, the generation strategy of *disjointWith* axioms may not be suitable for all the ontologies so that it could influence the performances of models. Hence, our experiments demonstrate that there is still much room for existing models to achieve schematic knowledge completion.

Nevertheless, there are issues worth discussing that have not yet been addressed in our current work. Notice that we assume that all the concepts in schematic knowledge or ontologies are static and correct. However, the axioms are usually updating in real scenarios. An incremental embedding method for schematic knowledge embedding should be explored, so as to avoid repeating training all over again whenever axioms update. On the other hand, if there are some wrong axioms asserted in ontologies without labeling, how to detect these wrong axioms and encode the correct ones at the same time becomes challenging. Therefore, one kind of embedding method that can be compatible with wrong axioms still needs to be considered.

## 7. Conclusions

In this paper, we presented a cosine-based embedding model called CosE for completing lightweight schematic knowledge in DL-Lite$_{core}$, by which the transitivity and symmetry of relations in axioms could be preserved simultaneously. To sufficiently learn the vector representation of concepts in axioms, we introduced the two semantic spaces and designed two types of score functions for them, which are tailored for axioms expressed in DL-Lite$_{core}$. Furthermore, we proposed a strategy of negative sampling derived from the mutual exclusion between *subclassOf* and *disjointWith* relations. In this way, CosE could learn better vector representations of concepts for completing schematic knowledge. We implemented and evaluated our model on four standard datasets generated using real ontologies. Experimental results have shown that CosE could simultaneously keep the logical properties (i.e., transitivity, symmetry) of relations and outperform state-of-the-art models in most cases.

For future works, we will study further along three directions: (1) CosE is an embedding model to complete the axioms of lightweight schematic knowledge, but it is limited to DL-Lite$_{core}$. It is worth extending CosE to more expressive ontology languages of schematic knowledge such as DL-Lite$_{\mathcal{A}}$ [51]. (2) Deep learning networks with the transformer architecture (e.g., BERT [52], GPT-3 [53], and their variants [54]) can be utilized to further optimize our model. All of them can take full advantage of large and external knowledge sources. Incorporating such models into our method can facilitate better performances being obtained for schematic knowledge completion. (3) The embedding of concepts could be helpful for the related tasks of ontologies and KGs. We will try to extend CosE so that it can be applied in these tasks and improve their performances, such as ontology-based data access [55], ontology matching [56], and knowledge graph refinement [57].

**Author Contributions:** Funding acquisition, W.L.; methodology, W.L. and H.G.; conceptualization, W.L. and H.G.; software, X.Z.; validation, X.Z.; writing, review and editing, Q.J. and G.Q.; supervision, G.Q. All authors have read and agreed to the published version of the manuscript.

**Funding:** This research was funded by the Natural Science Foundation of China, grants (62006125, U21A20488, and 61602259), and the Foundation of Jiangsu Provincial Double-Innovation Doctor Program, grant (JSSCBS20210532), and the NUPTSF grant (No. NY220171).

**Institutional Review Board Statement:** Not applicable.

**Informed Consent Statement:** Not applicable.

**Data Availability Statement:** Not applicable.

**Conflicts of Interest:** The authors declare no conflicts of interest.

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
