# Peer review of "Cosine-Based Embedding for Completing Lightweight Schematic Knowledge in DL-Litecore†"

_applsci, doi:10.3390/app122010690_

Round 1

Reviewer 1 Report

The authors have presented an interesting work. They have compared their work with the existing works. However, the authors should focus on the following comments.

1.       The authors should check the language. There are minor grammatical mistakes including punctuations. For example, “Schematic knowledge, as an important” should be “Schematic knowledge, an important”.

2.       The authors should avoid duplicate sentences in two different sections of the manuscript. The authors should avoid repeating the same information in more than one sentence.  

3.       The definition for “triples” is unclear.

Reviewer 2 Report

The authors tackles in their paper a solution for semantic research. The proposed method tries to deal with some limitations such as KG embedding methods pay more attention to encoding factual knowledge, but few of them consider the completion of schematic knowledge. And several methods try to preserve logical properties of relations for completing schematic knowledge, but most of them cannot simultaneously preserve the transitivity. To deal with two limitations the authors propose a cosine-based embedding model named CosE (Cosine-based Embedding) tailored for completing lightweight schematic knowledge in DL-Litecore. The authors produce promising results with proposed model.  

The authors presented an overview for the state of the art.

The contributions of the research are presented.

The paper in current state can be considered for publication, yet there are some changes need to be carried out for final version.

·         Abstract too long for a research paper, the author should give a brief information in order to present the paper.

·         The authors should produce a matching algorithm to help the reader understand the work.

·         What is the difference between CosE and CosE?

Reviewer 3 Report

In my opinion this paper crossed the line of autoplagiarism and the amount of novel content is not sufficient to justify a journal publication. In my view, 90% of the content is the same as in the paper "Cosine-Based Embedding for Completing Schematic Knowledge" from 2019 but reformulated, one example provided below. It should be the opposite, ie. 80% of content (including ideas) should be new.

The paper should clearly distinguish and emphasize what is the contribution of this work, and what has been already published in the 2019 paper. The content copied from the 2019 paper should be mostly removed, maybe except a brief explanation and a reference.

Sections 4.1 and 4.2 in this paper are a copy of Sections 3.1 and 3.2 in Gao et al., 2019. While you do not copy paste the text, the equations and ideas are the same, the text is only rephrased. I find it dishonest. One example:
"CosE has obviously surpassed other models in terms of MRR and Hits@N." (new paper)
"CosE significantly outperforms the models in term of Hits@N and MRR. " (old paper)
And all of the text is just like that - is it to prevent automated plagiarism detectors?

What I find perhaps novel is the content of lines 284-298 in Section 4.3 in the new paper. This is not enough to justify a new journal paper.
In all the tables in Section 5.2 of the new paper, you should provide reference in each row, if the presented results are a copy from [1] or some other published work, and clearly distinguish from novel results reported in this paper.
As to the novel content of Section 5.3, it is also problematic, maybe one bullet point at the end.

The same remark concerns Section 5, it should clearly describe the contribution of this paper compared to your old paper from 2019 and remove discussion and conclusions published there, but here only rephrased.

To sum up, if you really consider this for publication, please add much more novel ideas and experiments (what about using the newest neural networks, including transformer architectures) and remove content copied from old paper, but only reformulated.

Round 2

Reviewer 3 Report

Thank you for addressing my concerns.